# Long-term coastal openness variation and its impact on sediment grain- size distribution: a case study from the Baltic Sea

Wenxin Ning[1], Jing Tang[2,3], Helena L. Filipsson[1]

[1] Department of Geology, Lund University, Sölvegatan 12, SE-223 62 Lund, Sweden

[2] Terrestrial Ecology Section, Department of Biology, University of Copenhagen, Copenhagen Ø, DK-2100, Denmark

[3] Center for Permafrost (CENPERM), University of Copenhagen, Copenhagen K, DK-1350, Denmark

*Correspondence to*: Jing Tang (Jing.Tang@bio.ku.dk)

**Abstract.** We analysed the long-term variations in grain-size distribution in sediments from Gåsfjärden, a fjord-like inlet in the southwest Baltic Sea, and explored potential drivers of the recorded changes in the sediment grain-size data. Over the last 5.4 thousand years (ka) in the study region, the relative sea level decreased 17 m, which was caused by isostatic land uplift. As a consequence, Gåsfjärden was transformed from an open coastal setting to a semi-closed inlet surrounded by numerous small islands on the seaward side. To quantitatively estimate the morphological changes in Gåsfjärden over the investigated time period and to further link the changes to the grain-size distribution data, a digital elevation model (DEM)-based openness index was calculated. The largest values of the openness indices were found between 5.4 and 4.4 cal. ka BP, which indicates relatively high bottom water energy. During the same period, the highest sand content (~0.4%) and silt/clay ratio (~0.3) in the sediment sequence were also recorded. After 4.4 cal. ka BP, the average sand content was halved to ~0.2% and the silt/clay ratio showed a significant decreasing trend over the last 4 ka. These changes were found to be

associated with the gradual embayment of Gåsfjärden, as represented by the openness indices. The silt/clay ratios exhibited a delayed and relatively slower change compared with the sand content, which indicates different grain-size sediments responses differently to the changes in hydrodynamic energy. Our DEM-based coastal openness indices have proved to be a useful tool for interpreting the temporal dynamics of sedimentary grain-size.

## 1 Introduction

Sedimentary grain-size distribution provides important information regarding depositional conditions and has been widely analysed in both modern samples and sediment cores (e.g., Tanner, 1992; Yang et al., 2008; Virtasalo et al., 2014). Grain-size distribution is generally governed by sediment inputs and hydrodynamic energy conditions. The higher the energy conditions, the larger proportion of coarse grains (Dearing, 1997; Jönsson et al., 2005). Water depth, wind direction and strength, basin morphometry, and man-made constructions such as dam-building, could influence bottom water hydrodynamics and may lead to different characteristics in grain-size distributions. The Baltic Sea is connected with the North Sea through the narrow Danish Straits. Although tidal activity can strongly influence grain-size distribution in coastal regions, as shown by Zhang et al., (2002), the tidal amplitude recorded in the Baltic Sea is only a few centimetres (Ekman and Stigebrandt, 1990) and therefore its impact on sediment grain-size is not considered in this region. Instead, wind conditions and coastal morphometry are considered to be the most important factors that influencing the sedimentary grain-size distribution in the Baltic Sea coastal zone (Lehmann et al., 2002; Jönsson et al., 2005; Al-Hamdani and Reker, 2007).

During the Holocene, the Baltic Sea has experienced several stages modulated by global sea-level changes as well as isostatic land uplift (Björck, 1995; Andrén et al., 2011). As the Late Weichselian ice sheet retreated, land uplift during the deglaciation and the Holocene resulted in shoreline displacements in the coastal zones of the

Baltic Sea. A maximum of 60 m decline in relative sea level (RSL) has been recorded over the last six thousand years (ka) (Påsse and Andersson, 2005), leading to basin isolations and long-term changes in the coastal morphometry (Eronen et al., 2001). These changes in the coastal morphometry variations may potentially be linked with variations in grain-size distributions, as shown from a coastal inlet in the southwest of Baltic Sea

(Ning et al., 2016).

To examine the impact of coastal morphometry changes on grain-size distribution in a long-term perspective, quantifying the morphological changes, e.g., water depth and cross-section areas could be useful. Achieving this quantification is data-demanding, since it requires 1) high-resolution bathymetry data, 2) right cross-section area and 3) sedimentation rates. However, all these data are difficult to obtain and makes the quantification difficult.

Alternatively, through using digital elevation maps, Lindgren (2011) proposed a geographical information system (GIS)-based wave fetch index, named filter factor, to estimate coastal morphometry. The result of the quantified coastal morphometry was found to be significantly correlated with bottom water dynamics (Persson and Håkanson, 1995) and deep-water turnover time (Persson and Håkanson, 1996). There are also other GIS-based indices existing for describing coastal openness and wave exposure (Ekebom et al., 2003; Tolvanen and

Suominen, 2005) and these GIS-based methods have also been applied to investigate sediments grain-size distributions from lakes and coastal zones (Håkanson, 1977; Lindgren and Karlsson, 2011). However, these aforementioned indices are restricted to depict the modern coastal morphometry and have not yet been employed in a paleoenvironmental context.

In this study, we proposed an openness index using Digital Elevation Model (DEM) data and this approach could

provide an opportunity to estimate long-term coastal morphometry variations for the Holocene. Furthermore, we innovatively used the coastal openness index for grain-size data interpretations. The aim of the study is to: (1) present a method for quantifying openness changes in coastal region that experienced large relative sea level as

well as shoreline changes, and (2) link the estimated openness index with the long-term sediment grain-size distribution.

## 2 Materials and methods

### 2.1 Study area and digital elevation model

Gåsfjärden is a semi-enclosed fjord-like inlet located on the southeast Swedish Baltic Sea coast (Fig. 1a). It has a restricted water exchange with the open Baltic Sea through a narrow and shallow strait (~500 m wide, <20 m deep) in the east, which hinders sediment transport between the inlet and open waters. The surface area of Gåsfjärden is 22 km$^2$ and the mean and maximum water depths are 10 m and 51 m, respectively. The RSL has decreased by 17 m in the region over the last 5.4 ka as a result of isostatic land uplift (Fig. 2) and the present

land uplift rate is around 1.5 mm yr$^{-1}$ (Påsse & Andersson 2005). The catchment of Gåsfjärden is characterised by very thin soils (< 1 m) and exposed pre-Cambrian bedrock. Arable land is sparsely distributed in the lowlands and vegetation mainly consists of coniferous forest (Fig. 1b). Small-scale human activities existed in the region as early as 2 ka ago, although substantial expansion has occurred since the 1700s (Karlsson et al., 2015; Ning et al., 2016). In the shallow waters of the inlet, sandy patches can be found in addition to the rocky coast. The

sediment accumulating in the inlet most likely originates from the terrestrial environment through erosion, and from land-run and river transport, instead from the open Baltic Sea. Sediment accumulation rate over the last 1000 years is generally less than 1.5 mm yr$^{-1}$ in the deep basin (Ning et al., 2016).

The Light Detection And Ranging (LiDAR)-based DEM data of the study region (Fig. 1c) were obtained from the Swedish mapping agency, Lantmäteriet (http://www.lantmateriet.se/). The horizontal and vertical resolutions

of the DEMs are approximately 2 m and 0.1 m, respectively. The data are in the Swedish national coordinate system (SWEREF99 TM).

**2.2 Chronology and grain-size analysis**

Sediment cores were collected at Station VG31 (57°34'21.3" N, 16°34'58.4" E) in August 2011 on the R/V *Ocean Surveyor*. A 6 m sediment sequence was obtained and the age-depth model was established through a combination of $^{210}$Pb, $^{137}$Cs and AMS-$^{14}$C dating methods (Ning et al., 2016). For the grain-size analysis, organic carbon, calcium carbonate, and biogenic silica were removed from the sediment samples using procedures by Van Hengstum et al. (2007). To obtain enough minerogenic material, a mixed sediment sample of about 13 g, with core sections of a maximum of 7 cm (covering ~60 years), was used. The sand particles (>63 µm) were sieved, dried and weighed. The mass fraction of sand was calculated by dividing the dried sand weight with the original dry sample weight before any chemical treatment. The mass fraction of clay (<2 µm) and silt (2-63 µm) from particles less than 63 µm were obtained with a Micromeritics Sedigraph III Particle Size Analyser at the Department of Geology, Lund University, Sweden.

**2.3 Openness index**

The calculation of the openness index (Fig. 3) has been modified on the basis of the method described by Lindgren (2011) and the fetch-length method of Ekebom et al. (2003). The following steps for estimating openness index variations were taken in ArcGIS 10.3:

1) The coring site was identified in the DEM.

2) Using the coring site as starting point, two sets of 180° circles of radiating lines were created. One set of radial lines was towards the east (seaward) and the other was towards the west (landward), with an interval of Δ degree (generated by the Python scripts in Supplement S1). The length of the radiating lines was set as 8 km and the interval Δ was set as 1-5, 10 and 15°. The radial lines of 8 km were used as they can reach offshore open water. The radial lines with a 5° interval but with different shifting angles θ (1-4°) were also created (Fig. 3a).

3) The RSL changes with a 100 years interval were applied to the present day DEM. For every 100 years, a new DEM was generated and the RSL changes were based on the age-RSL relationship in Fig. 2.

4) For every 100 years, the grid cells in the generated DEM were classified as sea or land based on the elevation.

5) The raster DEMs were converted to land and sea polygons for vector calculation in ArcGIS, and the radiating lines generated in Step 2) were divided into smaller segments when the lines were intersected by the land polygons.

6) The lines originating from the coring site and that came into contact with the land were selected (see Figs. 4 and 5). The seaward and landward openness indices were calculated as the average length of the selected radiating lines.

## 3 Results and Discussion

### 3.1 Estimated openness indices under different scenarios

The openness indices with different intervals have been estimated in order to determine an optimal interval for applying this index for the study area. The seaward and landward openness indices, calculated with 15° and 10° intervals, exhibited relatively large year-to-year deviations, compared with the indices using the relatively smaller degree intervals (Fig. 6). The calculated landward and seaward indices using the 15° interval are at the maximum 7 % and 20 % larger than the 1° interval scenario. Only minor differences (maximum 5 %) were observed between the openness indices calculated using 1, 2 and 3° intervals. Generally, the larger degree of intervals, such as 5°, 10° and 15°, resulted in fewer radial lines; and as a consequence, the weights of a few very long or short radial lines on the average length will be relatively larger compared with the estimation using smaller intervals. Therefore, the associated uncertainties in the estimated openness index will be larger when

using radial lines with larger intervals. The high frequency of radial lines, e.g., using a 1° interval, ensures a higher possibility for capturing the details of the coastal openness variability. Furthermore, potential effects of shifting angles (angle between the north and the nearest radial line) were tested and Figure 7 showed the cases for the 5° interval of radial lines with shifting angles of 0° to 4°. The results demonstrated that using different shifting angles can cause substantial differences in the estimated openness indices when the radial intervals are relatively large. However, if the interval is set as 1°, changing the shifting angle from 0° to 4° would result in openness indices with very small differences in consideration of relative changes in the positions of all radial lines. Therefore, using low degree interval such as 1° for calculating the openness indices is preferred and should be recommended for other similar studies, although the computing time would be longer than higher degree intervals. The landward and seaward openness indices were differentiated although they both reflect morphological changes of the inlet over the last 5.4 ka. The seaward openness index reflected more accurately the embayment process in comparison with the landward openness index, as the most distinct changes of the inlet was from the sill in the east. The landward openness index, reflecting offshore distance, could also be important for considering sedimentary grain size. Seaward and landward openness indices have demonstrated a continuous decline over the last 5.4 ka (Fig. 6), caused by isostatic land uplift and the embayment process. The decreasing rate of the seaward openness index was more pronounced during 5.4-4.4 cal. ka BP than the period of 4.4-0.1 cal. ka BP. In contrast, the decline in the landward openness index was relatively smoother and no drastic transition was recorded ~4.4 cal. ka BP.

**3.2 Implications for sediment grain-size distributions**

The openness indices calculated with 1° interval and 0° shifting angle were plotted along with the grain size data (Fig. 8). The correlations ($r^2$) between the openness indices and grain size data range between 0.47 and 0.65 (p < 0.01) (Table 1). The significant correction suggested that the change in coastal openness was an important factor

influencing the sedimentary grain size distribution. Both the seaward openness index and sand content had the highest values between 5.4 and 4.4 cal. ka BP, and there is a synchronous large decline in seaward openness and sand content around 4.4 cal. ka BP (Fig. 8). This further indicates a connection between the depositional environment and coastal openness variations. The maximum sand content at the coring site was only ~0.4%,

suggesting a relatively low bottom water velocity compared with open Baltic Sea waters (Jönsson et al., 2005). Generally, in coastal zones, the closer to the shore, the more coarse-grained sediments can be deposited. However, the sediments in Gåsfjärden became more and more fine-grained as the coring site became shallower and closer to the shore (closer to present time, see Figs. 5 and 8), which was a result of less exposure and an increasingly protected location. At present, the sea-floor outside Gåsfjärden is characterised by sandy sediments,

whereas gyttja clay is deposited in the sheltered Gåsfjärden (Al-Hamdani and Reker, 2007). The modern grain-size difference between the areas inside and outside Gåsfjärden is linked to the different hydrodynamic statuses. The relatively higher sand content during 5.4 and 4.4 cal. ka BP may also be associated with the increased sand transport when Gåsfjärden had a relatively larger cross-sectional area. Based on an analysis of 201 sites along the Swedish coast characterised by complex bathymetry, Lindgren and Karlsson (2011) estimated that the mean

critical depth separating the depositional areas from the erosion and transport areas is located at 19 m. At present, our coring site has a water depth of 31 m. The relatively deeper water depth (sediment dominated by transport, instead of erosion) together with sheltered condition resulted in the low sand content.

Erosion from the surrounding islands since 5.4 ka has most likely occurred, but it has been rather limited as these islands are mostly rocky with little soil cover. It may, however, result in a flux of relatively coarser grains to the

coring site during the land-uplifting. As the uplift process has shown to be linear (see Fig. 2), we might expect to see a rather linear change in the grain size data, if the land-uplifting had played the dominant role. However, the sand content and silt/clay ratios exhibit strong year-to-year variations, which indicates other factors than land uplifting could also participate in influencing grain-size distribution. For instance, coarse grains, such as sand,

can also be transported to the coring site through storm events and intense wave action, sea ice or drifting sea weed. However, their impacts are not explicitly included in the openness indices. Furthermore, the recorded large variability in the sand content within the last millennium may be linked with catchment disturbance from human activities (Karlsson et al., 2015; Ning et al., 2016).

The silt/clay ratios also reflected bottom water energy, with higher values indicating higher energy conditions. The silt/clay ratio was ~0.3 between 5.4 and 4.0 cal. ka BP, and exhibited a continuous decline from 4.0 cal. ka BP. The different pattern between the silt/clay and sand content, particularly during the period when Gåsfjärden was relatively open, between 5.4 and 4.0 cal. ka BP, suggested that different grain-size sediments respond differently to the changes in hydrodynamic conditions in this region. The decrease of sand content from 5.4 to

4.4 cal. ka BP indicates a decline in bottom water velocity. However, the bottom water velocity was probably still high enough to maintain the silt/clay ratios, as silt and clay might have responded similarly to the changes in hydrodynamics. The different grain-size classes respond differently to hydrodynamic changes were also reported in two coastal sites in Italy (Molinaroli et al., 2009), where positive correlations between current velocities and silt (8-63 μm) and fine sand (63-105 μm) fractions were found in the two lagoons. The changes in landward

openness, which reflect the offshore distance, seem to follow the silt/clay ratios ($r^2 = 0.65$). In our data, the silt/clay ratios exhibited a significant decreasing trend (p<0.01, Mann-Kendall test) between 4.0 and 0.1 cal. ka BP, which indicates a long-term impact of decreased coastal openness on the grain-size distributions.

**4 Conclusions**

Our DEM-based calculations of coastal openness indices have shown to be a useful tool when interpreting long-

term sedimentary grain-size data. A relatively high relative sea level was linked with large coastal openness and higher hydrodynamic energy, which in turn was well reflected in the seaward openness index. The higher values of both sand content and the seaward openness index were recorded in the early part of the record, indicating that

coastal morphology (presented by openness index) strongly influenced sand distribution. The differences in temporal dynamics of sand content and silt/clay ratios indicate different grain-size sediments responds differently to the changes in hydrodynamic energy. The significant decline in silt/clay ratios between 4 and 0.1 ka demonstrated that the long-term impacts of coastal openness on the finer grain-size sediment distributions. Our DEM-based openness index can be easily applied to other coastal settings that have experienced large sea-level changes over time. The index could also be further used in predicting future dynamics by combing information about sea-level changes in a warmer future.

**Acknowledgements**

Z. Wu is thanked for the help on data analysis. The project was funded by FORMAS Strong Research Environment: Managing Multiple Stressors in the Baltic Sea (217-2010-126). We thank the captain and crew of R/V *Ocean Surveyor* for help during sampling. We thank N. V. Putten and Å Wallin for guidance during grain-size analysis. We also acknowledge funding from the Crafoord Foundation and the Royal Physiographic Society in Lund. J. Tang was financed by the Villum foundation (VKR022589) and The Danish National Research Foundation (CENPERM DNRF100). We thank Dr. Evan Goldstein and another anonymous reviewer for their helpful comments on the manuscript.

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

**Tables and Figures**

Table 1. Correlation ($r^2$, $p < 0.01$) matrix for the grain size data and the calculated openness indices

| Index | Sand | Silt/clay |
|---|---|---|
| Landward openness | 0.48 | 0.65 |
| Seaward openness | 0.47 | 0.56 |

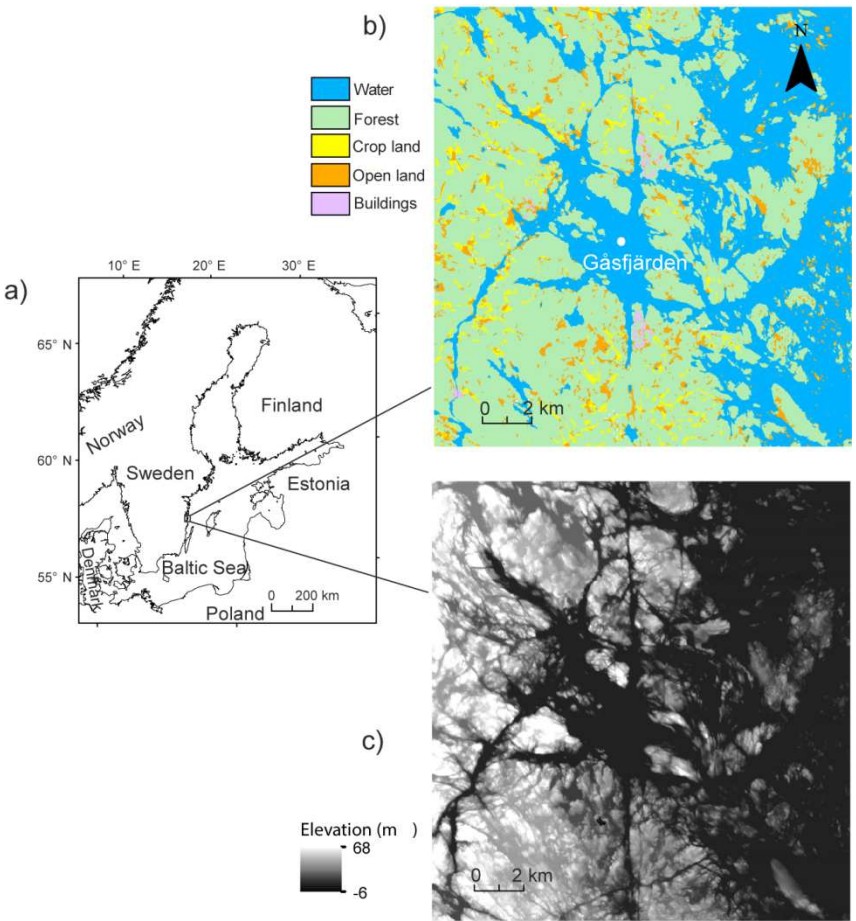

**Figure 1 (a) Overview of the Baltic Sea region and the location of Gåsfjärden; (b) the coring site (white filled circle) and land use map, (c) digital elevation model of the study region.**

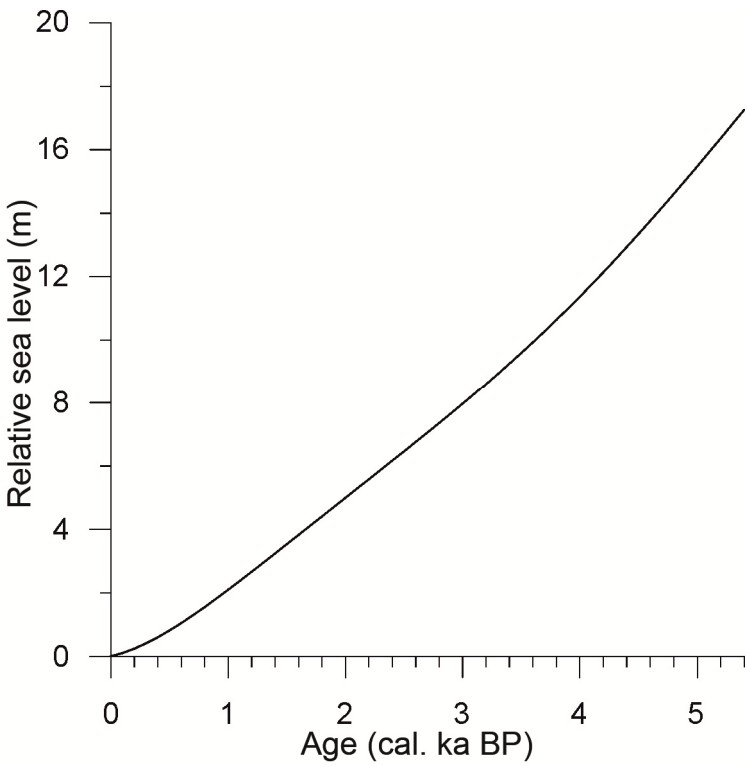

**Figure 2 Changes of relative sea level in the study area over the last 5.4 ka based on empirical model by Påsse and Andersson (2005).**

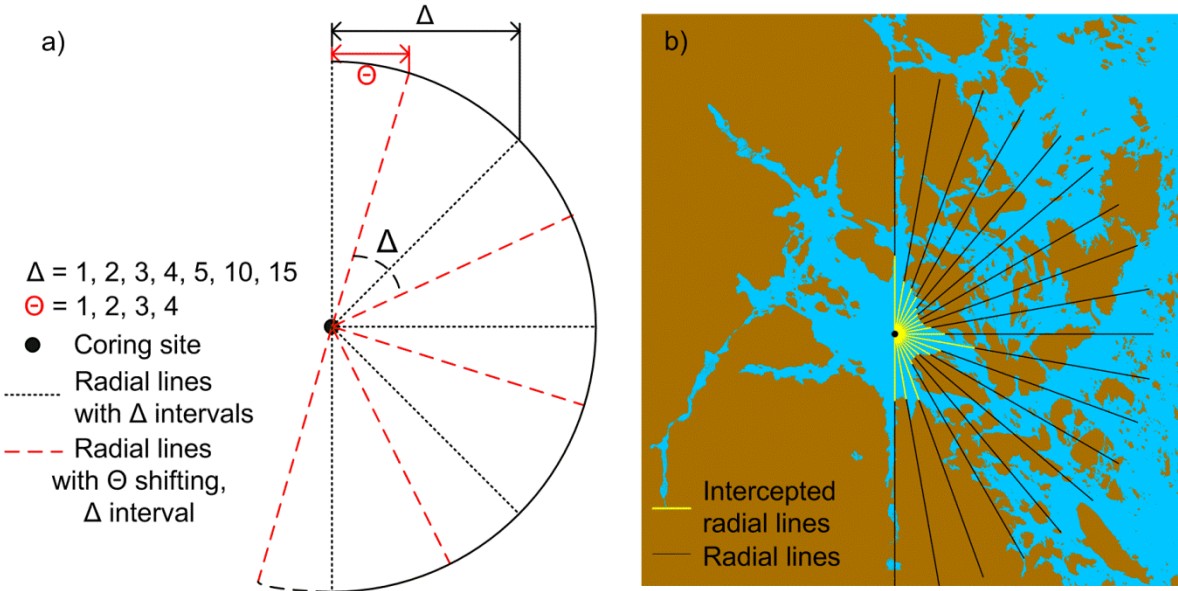

**Figure 3 (a) Illustration of the 180 degree radiating lines, the intervals (Δ), and the shifting angles (Θ); (b) black and yellow lines representing the 8 km-long radial lines and the intercepted lines for openness index calculation.**

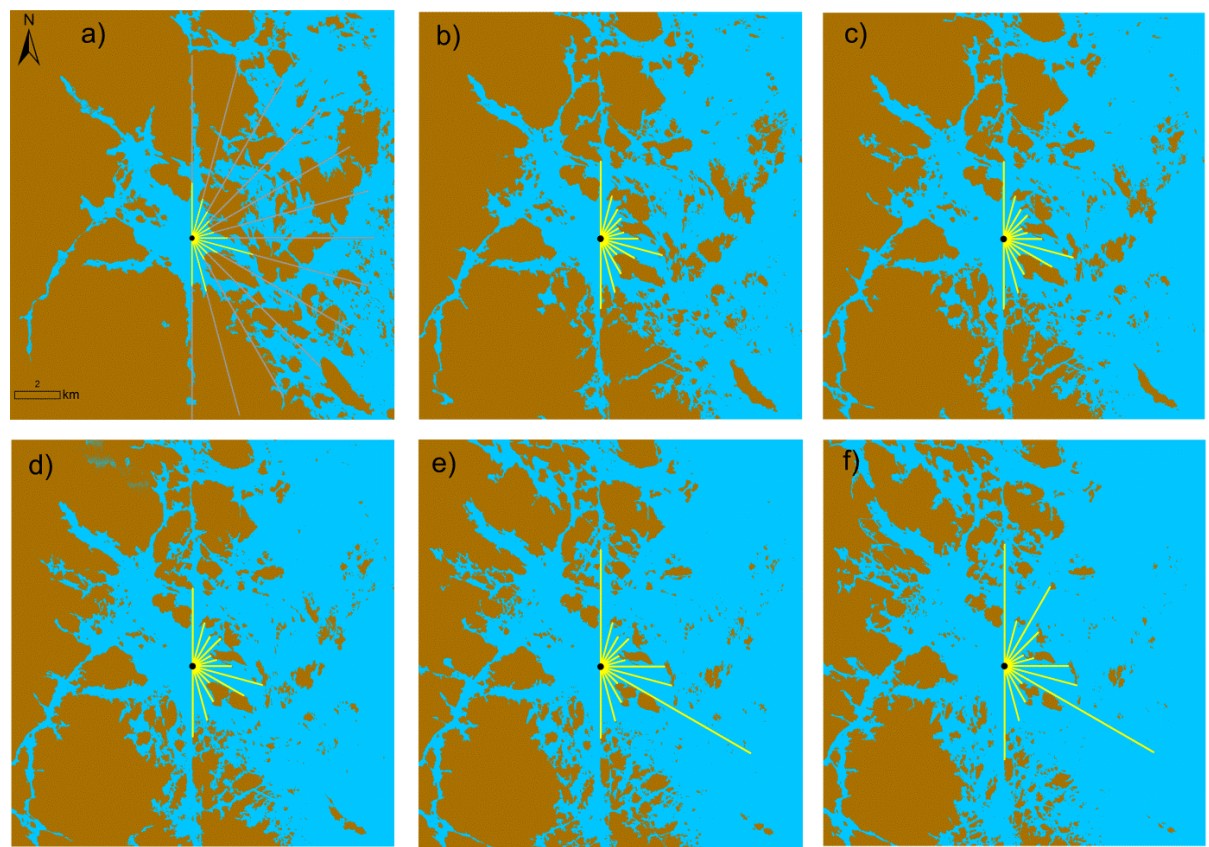

**Figure 4 Illustrations showing the variations of seaward radial lines intercepted with land over the last 5.4 ka. (a) 0.1 cal. ka BP (b) 2.5 cal. ka BP (c) 3.5 cal. ka BP (d) 4 cal. ka BP (e) 4.5 cal. ka BP (f) 5.4 cal. ka BP.**

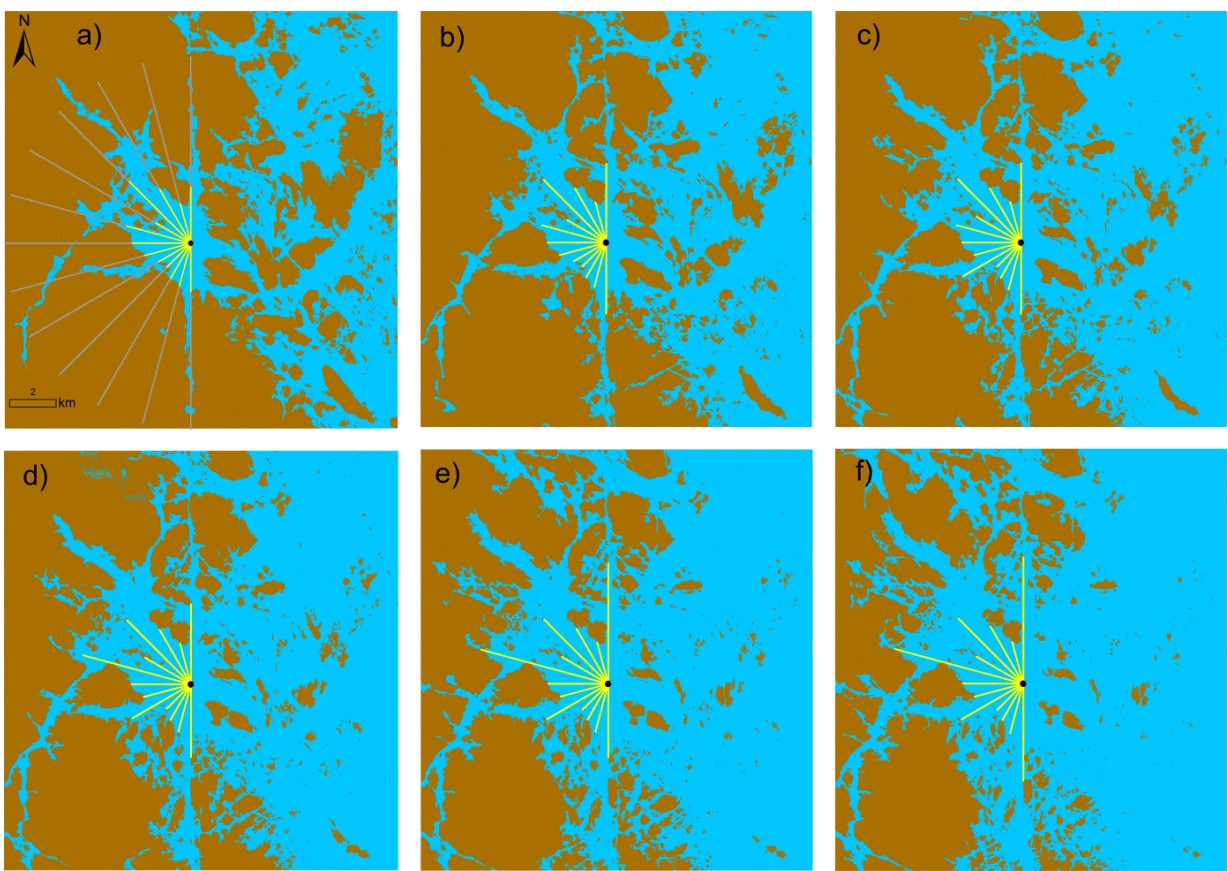

**Figure 5 Illustrations showing the variations of landward radial lines intercepted with land over the last 5.4 ka. (a) 0.1 cal. ka BP (b) 2.5 cal. ka BP (c) 3.5 cal. ka BP (d) 4 cal. ka BP (e) 4.5 cal. ka BP (f) 5.4 cal. ka BP.**

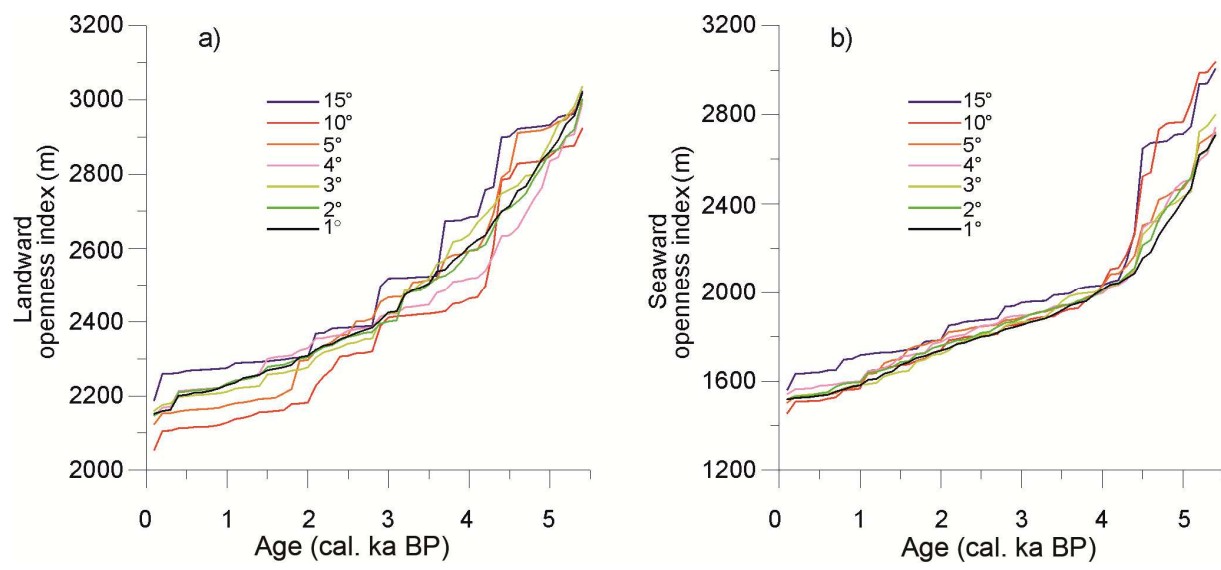

**Figure 6 Calculated landward (a) and seaward (b) openness indices of Gåsfjärden over the last 5.4 ka, using 1-5, 10 and 15° intervals.**

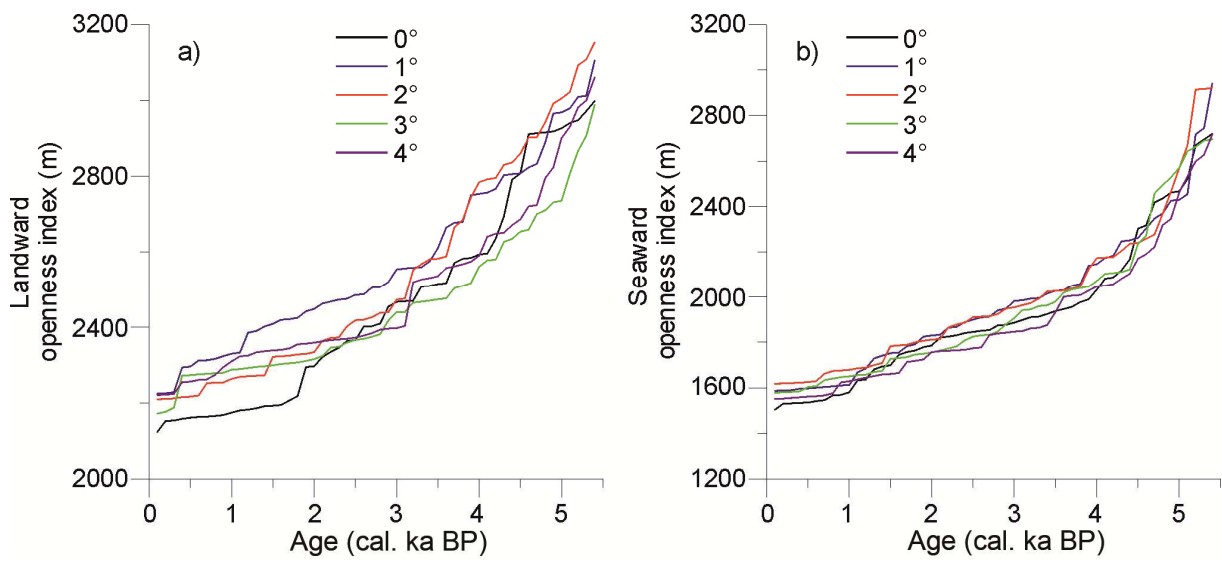

**Figure 7 Calculated landward (a) and seaward (b) openness indices with 5° interval and 1-4° shift starting angles.**

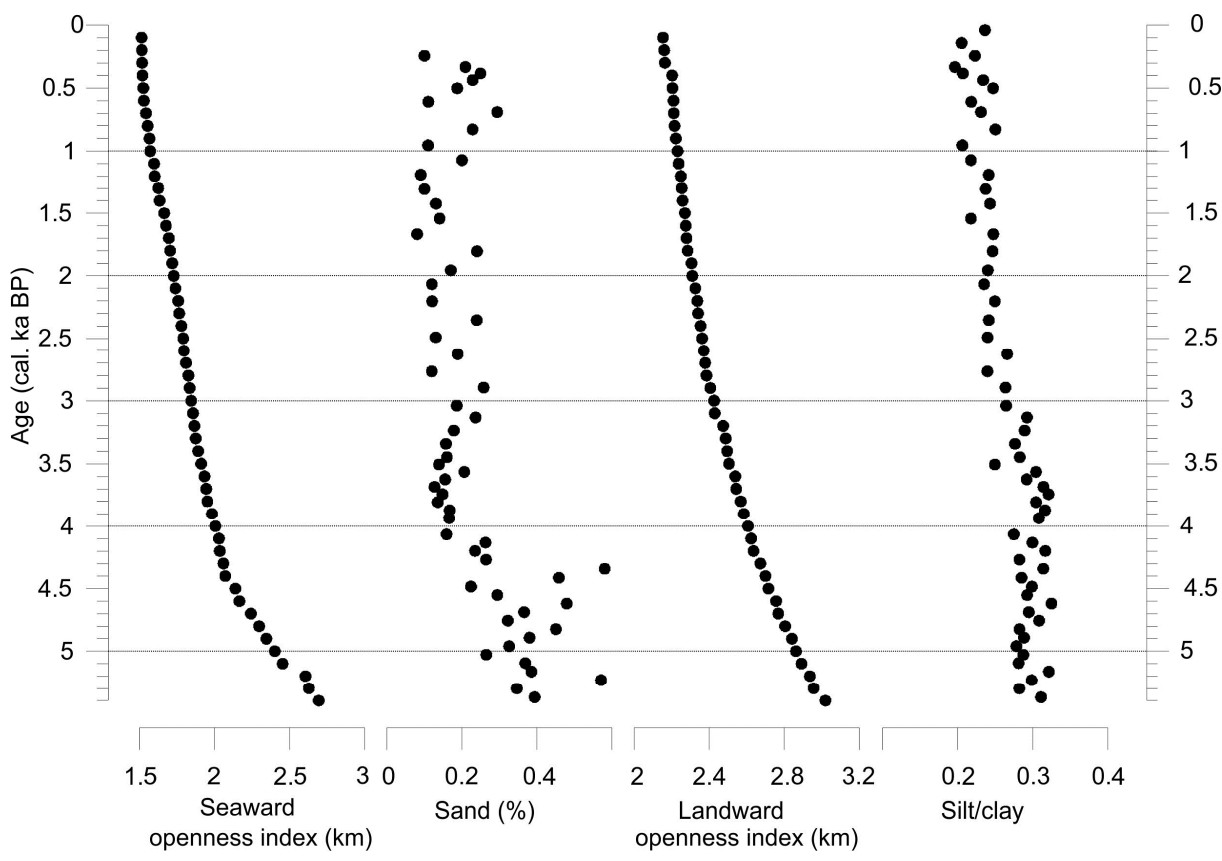

**Figure 8 The calculated openness indices using 1° interval and 0° shifting) and the measured grain size from the sediment.**

