# Peer review of "Long-term coastal openness variation and its impact on sediment grain- size distribution: a case study from the Baltic Sea"

_Earth Surface Dynamics, 2016_

## Referee Comment (RC1) · E. Goldstein (Referee) · 5 May 2016

Ning, Tang and Filipsson use GIS data coupled with grain size data from core to analyze the effect of sea level changes on the sedimentary record of an inlet system.

I have several major and minor comments, listed below.

Respectfully, Evan B Goldstein

Major Comments:

1) I believe this manuscript could benefit with more description as to the mechanics of sediment transport in this specific system to justify the results (Section 3.2). For

instance, what drives sand transport in the modern system? Does sand come from the Baltic into the inlet? or is the sand coming from the terrestrial setting? i.e., as a reader it would be helpful to understand in more detail how this physical system works?

2) Can the authors connect openness index with a near bottom water velocity and sediment transport in some way - i.e., fetch, wind speed, and water depth to calculate wave orbital motions at the bed using the relations presented in Young and Verhagen (1996)? Or perhaps the authors could relate the (spatial) change in openness index to the wind field (modern or ancient) and the fetch?

3) The authors focus on developing an 'openness index' which is the average length of line from the core site to land at a given time/sea level. Why are landward vs. seaward openness indices differentiated? And a related comment, the shifting angle is discussed only briefly. Can the authors give us some guidance on picking a starting position? Do any radial lines, at any time, make it to the open Baltic sea (i.e., do any openness measurements exceed the 8 km line segments used)? Are these lines important? (i would presume so, because these directions would permit larger waves into the system and exert more work on the bed.)

4) The authors present Figure 6 and 7 to show there is variation in the openness index for a given degree interval (or shifting angle) at a given time. Is there a way to make this analysis more quantitative? (i.e., p5, line 9; how much 'larger'?) One suggestion to illustrate this in the figures is to plot openness variance as opposed to the raw openness index. On a related note, the authors state that they endeavor to find an optimal degree interval (p. 5 line 3). I assume 'optimal' in this context refers to a negligible variance in openness index relative to decrease computation time (associated with increasing the degree interval)? Perhaps quantifying the variation in openness index for a given degree interval will aid them in searching for an 'optimal' interval?

5) The authors present openness index data and grain size in figure 8. I believe more quantitative analysis could be performed with this data to convince the readers. For

instance, what values of shifting angle and degree interval was used? why? What is the correlation between opening index vs sand %? or openness index vs silt/clay?

6) Has there been erosion of the islands since 5 ka? (i.e., is the present subaerial expression of the islands identical to the coastlines of the island in the past?) how could this impact your study?

Minor Comments:

-The manuscript referee to 'radical' lines (e.g., p4; line 10-15). Would 'radial' be a more fit word instead?

-The reference for Al-Hamdani and Reker is incomplete.

---

## Referee Comment (RC2) · Anonymous Referee #2 · 10 May 2016

Comments to Wenxin Ning, Jing Tang & Helena L. Filipsson: Long-term coastal openness variation and its impact on sediment grain-size distribution: a case study from the Baltic Sea

This is a well-written and well-illustrated manuscript that is suitable for Earth Surface Dynamics

However, I would like to see a few more notes about the setting: are there only rocky coasts, or are there also patches af sandy shores? And what about shallow waters? All rocks? Some notes are found in 3.2, but more notes could be added to 2.1.

I also wonder how sand is transported to the core site. Does it happen during storms

as storm sand layers? Is sand blown out on the sea ice during cold winters? Is sand transported by drifting sea weed or by drifting sea ice?

I would also like to see a few notes on the chronology of the core, at least a reference to Ning et al. (2016).

The main control on grain size distribution is distance to the shore, but this is apparently not mentioned. The closer to the shore – the more coarse-grained sediments. In Gåsfjärden, however, the sediments become more and more fine grained as the core site moves closer to the shore. This is not surprising, because the core site at the same time becomes more and more protected. The authors have developed a novel GIS-based approach that allows them to quantify down-core changes in grain size distributions in relation to changing fetch.

---

## Short Comment (SC1) · 31 May 2016

Comments from Evan Goldstein and our replies: 1) I believe this manuscript could benefit with more description as to the mechanics of sediment transport in this specific system to justify the results (Section 3.2). For instance, what drives sand transport in the modern system? Does sand come from the Baltic into the inlet? Or is the sand coming from the terrestrial setting? i.e., as a reader it would be helpful to understand in more detail how this physical system works?

Reply: The catchment of the inlet is characterized with thin soil (<2m) and the inlet has only small rivers draining. Therefore we speculate that sediment composition from the coring site is mostly governed by internal sediment redistribution and terrigenous

input. Sediment transportation from outside the inlet possibly also contributed to the sediment accumulation to a smaller degree. However, the narrow and shallow connection between the inlet and the open Baltic Sea may restrict the sediment transportation. Over the last 1 ka, sand content in the inlet was apparently influenced by catchment disturbance. This is supported by elevated sand content together with increased regional land use intensity during the last 1 ka (Karlsson et al., 2015). Internal distribution also influences grain size. For instance, strong wind or storm events may transport sand from near-shore depth to deeper off-shore areas. During the beginning part of the record around 5.4 ka, relatively high openness has caused an elevated energy environment and relatively high sand contents. The sand content is overall low through the record, linked with the lack of source and low energy status of the inlet.

Karlsson, J., Segerström, U., Berg, A., Mattielli, N., and Bindler, R.: Tracing modern environmental conditions to their roots in early mining, metallurgy, and settlement in Gladhammar, southeast Sweden: Vegetation and pollution history outside the traditional Bergslagen mining region, The Holocene, 25, 944-955, 10.1177/0959683615574586, 2015.

2) Can the authors connect openness index with a near bottom water velocity and sediment transport in some way - i.e., fetch, wind speed, and water depth to calculate wave orbital motions at the bed using the relations presented in Young and Verhagen (1996)? Or perhaps the authors could relate the (spatial) change in openness index to the wind field (modern or ancient) and the fetch?

Reply: Thanks for these interesting ideas. However, there is a lack of reliable reconstructions on historic wind speed and direction in the Baltic Sea region. Thus, calculating the velocity or fetch in the inlet is not possible at this moment. In future, with reliable long-term scale wind speed and water depth data available, linking the openness index, together with transport velocity and wave motion to the sediment transport is of great interest. The spatial change in the openness can be interesting to compare with grain size changes in the inlet. However, due to the lack of grain-size distribution

in the inlet, such comparison cannot be achieved so far. We therefore have focused on the coring site, from where we have both the temporal grain size data and variations in the openness.

3) The authors focus on developing an 'openness index' which is the average length of line from the core site to land at a given time/sea level. Why are landward vs. seaward openness indices differentiated? And a related comment, the shifting angle is discussed only briefly. Can the authors give us some guidance on picking a starting position? Do any radial lines, at any time, make it to the open Baltic sea (i.e., do any openness measurements exceed the 8 km line segments used)? Are these lines important? (i would presume so, because these directions would permit larger waves into the system and exert more work on the bed.)

Reply: The seaward openness index is the most important factor that governs the wave energy in the inlet. High wave energy in an open system leads to larger grain size. The landward openness reflects mostly changes in water depth. Lowered water depth would lead to larger grain size in an enclosed system. In the study site, as water depth decreases, the grain size also decreases. This indicates that seaward openness is more important in driving the grain size changes. The maximum 8 km lines were used because they have reached open water region for scenarios covering the last 5.4 ka and were recognized as a reasonable limit. Longer radical lines may result in relative higher values of openness index when the sea is open. For most cases, the radial lines have already intersected with islands at less than 8 km distance from the coring site (see Fig. 4). Therefore, we think the derived pattern of temporal variations in the openness indices should be similar and/or comparable with the current 8 km scenarios. These segments beyond 8 km link are considered to be small portions (see Fig. 4f) even in 5.4 ka scenarios. This will be further clarified in the revised version.

4) The authors present Figure 6 and 7 to show there is variation in the openness index for a given degree interval (or shifting angle) at a given time. Is there a way to make this analysis more quantitative? (i.e., p5, line 9; how much 'larger'?) One suggestion to

illustrate this in the figures is to plot openness variance as opposed to the raw openness index. On a related note, the authors state that they endeavor to find an optimal degree interval (p. 5 line 3). I assume 'optimal' in this context refers to a negligible variance in openness index relative to decrease computation time (associated with increasing the degree interval)? Perhaps quantifying the variation in openness index for a given degree interval will aid them in searching for an 'optimal' interval?

Reply: Different shifting angles and intervals are used to test if there is large difference among them. The results in Fig. 6 and 7 demonstrate that there are large variances among different shifting angles and intervals. To plot variances compared with shifting angle of 0 and degree interval of 1 might be a good way to illustrate the differences. But this will lead to unknown pattern of the raw openness index. Therefore we think it is good to keep Fig. 6 and 7 and add two figures on variance. The figures will be added in the revised manuscript. We think the raw openness index here means shifting angle of 0 and degree interval of 1.

5) The authors present openness index data and grain size in figure 8. I believe more quantitative analysis could be performed with this data to convince the readers. For instance, what values of shifting angle and degree interval was used? Why? What is the correlation between opening index vs sand %? or openness index vs silt/clay?

Reply: Shifting angle of 0 and degree interval of 1 are used in the figure 8 scenarios. The values were chosen because they tend to result in the most representative openness index, as discussed in 3.1 and the reply above. The correlation can be calculated to convince the readers and the coefficient matrix is listed below.

Correlation matrix for the grain size data and the calculated openness indexes Sand Silt/Clay Landward openness 0.48 0.65 Seaward openness 0.47 0.56

6) Has there been erosion of the islands since 5 ka? (i.e., is the present subaerial expression of the islands identical to the coastlines of the island in the past?) how could this impact your study?

**ESurfD**
Reply: Erosion from the island is most likely weak as these islands are mostly rocky with very thin soil. We cannot exclude that erosion has brought larger grains in size into the system during the land-uplift process, but the magnitude of the impact is difficult to evaluate. If there is continuous erosion since 5 ka, the sand contents and the silt/clay ratio would be more stable. This indirectly suggests that the impact of erosion from the islands is rather limited.

---

## Short Comment (SC2) · 31 May 2016

Reviewer's comments: I would like to see a few more notes about the setting: are there only rocky coasts, or are there also patches af sandy shores? And what about shallow waters? All rocks? Some notes are found in 3.2, but more notes could be added to 2.1. I also wonder how sand is transported to the core site. Does it happen during storms as storm sand layers? Is sand blown out on the sea ice during cold winters? Is sand transported by drifting sea weed or by drifting sea ice? I would also like to see a few notes on the chronology of the core, at least a reference to Ning et al. (2016). The main control on grain size distribution is distance to the shore, but this is apparently not mentioned. The closer to the shore – the more coarse-grained

sediments. In Gåsfjärden, however, the sediments become more and more fine grained as the core site moves closer to the shore. This is not surprising, because the core site at the same time becomes more and more protected. The authors have developed a novel GIS-based approach that allows them to quantify down-core changes in grain size distributions in relation to changing fetch.

Reply: The coastal region is characterized with rocky coasts, with sandy patches in the offshore water. Inside the inlet, there is so far no data about spatial sediment grain size distribution. Even so, the sand content in the inlet must be relatively low, due to the lack of large rivers and enclosed setting. The sand content is generally lower than 1% and we cannot presently determine how important storm events or sea ice are for the transport and abundance. During periods with relatively high openness, storm events would most likely transport higher amount of sand into the inlet. The chronology of the core will be added. We agree with the argument from the reviewer on the water depth-grain size relationship.

---

## Author Comment (AC1) · 1 Jul 2016

The authors would like to thank reviewer Evan Goldstein for giving the constructive comments, which will definitely improve the manuscript. Below we firstly addressed each comment and also indicate changes that have been made in the manuscript.

Comment #1: I believe this manuscript could benefit with more description as to the mechanics of sediment transport in this specific system to justify the results (Section 3.2). For instance, what drives sand transport in the modern system? Does sand come from the Baltic into the inlet? Or is the sand coming from the terrestrial setting? i.e., as a reader it would be helpful to understand in more detail how this physical system works?

Authors' reply on comment #1: Thanks for the good points. As the catchment of the inlet is characterized with thin soil and the inlet only has a few small rivers draining into it. On one hand, we speculated that sand transportation into the inlet from the catchment could be limited. On the other hand, the sand input into the inlet from the offshore regions could be very small, as a result of the narrow and shallow sill between the inlet and the open sea water. Overall, sand and/or sediment can be transported into the inlet through both terrestrial input and offshore region with limited amount (supported by relatively low sedimentation rate, <1.5 mm yr-1, during the last 1 ka). We speculated that sediment accumulated in the inlet mostly originates from the terrestrial setting, compared with the sandy offshore region.

Changes made in the manuscript based on comment #1: Page 3, line 9: "which hinders sediment transportation between the inlet and the open water." is added after the sentence "It has a restricted water exchange with the open Baltic Sea through a narrow and shallow strait (500 m wide, <20 m deep) in the east".

Page 3, line 15: "Therefore, there is a lack of erodible soil and subsequent sediment transportation into the inlet. Even so, sediment accumulated in the inlet is expected to originate mostly from the terrestrial setting, compared with sediment transportation from the sandy offshore region. Sediment accumulation rate over the last 1 ka is generally less than 1.5 mm per year (Ning et al., 2016)." is added after the sentence "The RSL has decreased by 17 m in…1.5 mm yr-1".

Comment #2: Can the authors connect openness index with a near bottom water velocity and sediment transport in some way - i.e., fetch, wind speed, and water depth to calculate wave orbital motions at the bed using the relations presented in Young and Verhagen (1996)? Or perhaps the authors could relate the (spatial) change in openness index to the wind field (modern or ancient) and the fetch?

Author's Reply on comment #2: Thanks for the great point. It would be really interesting to relate the estimated openness index with other environmental variables to potentially

explain sediment transport mechanism. However, there is a lack of reliable data on historic wind speed and direction in the Baltic Sea region. Thus, calculating wave orbital motions at the bed for the long-term scale is not possible at this moment.

As the reviewer correctly pointed out, it could be interesting to link the spatial change in the openness to the changes of wind field in the inlet, which may improve the ability of our current method in explaining sediment variation. However, since the main focus of this study is to explain the temporal dynamics of grain size changes at long-term scale, it may bring additional uncertainties to our estimations if we only use the available modern wind data for the past 5.4 ka. With availability of reliable wind data at long-term scale in future, it will definitely be interesting to explore potential impacts of wind on the grain size changes at the core site.

Changes made in the manuscript based on comment #2: No changes have been made.

Comment #3: The authors focus on developing an 'openness index' which is the average length of line from the core site to land at a given time/sea level. Why are landward vs. seaward openness indices differentiated? And a related comment, the shifting angle is discussed only briefly. Can the authors give us some guidance on picking a starting position? Do any radial lines, at any time, make it to the open Baltic sea (i.e., do any openness measurements exceed the 8 km line segments used)? Are these lines important? (I would presume so, because these directions would permit larger waves into the system and exert more work on the bed.)

Author's reply on comment #3: Both the seaward and landward openness indices can be linked with fetch and wave energy in the inlet, where high indices values potentially indicate relatively large bottom velocity. Thus higher openness indices in the open system lead to larger grain size in the sedimentation area. In comparison with the landward openness, the seaward openness index better reflects the morphological changes of the inlet, which is the main cause for hydrodynamic energy changes in the inlet over the last 5.4 ka. The landward index is used to describe the changes in

offshore distances and it can be important if prevailing wind direction is from the land to the sea. Thanks for the suggestions on adding comments on the shifting angle. The shifting angles of 0° to 4° have been used to test whether different starting angles influence the openness indices. The results presented in Fig. 7 showed that using 5° interval and different shifting angles, the changes in openness indices were substantial. If the interval is set as 1°, changing the shifting angle from 0° to 4° would be lead to little changes in the openness indices. Therefore using low degree interval such as 1° for calculating the openness indices is preferred. The optimal interval for estimating openness index could vary from different coastal settings and we suggest to test it before apply the index with other proxy data. In our study, using 1° interval would give the most robust results when calculating openness indices, although the computing time would be longer than larger degree intervals.

We have sediment data from the core site and this is also the site we are interested to investigate factors impacting sedimentation process. So it is straightforward for us to use the core site as the starting point of radial lines and the estimated changes of openness index can further link the index with other measured sediment variables.

The maximum length of 8 km line was used because it reached open water region for scenarios at the 5.4 ka ago and is recognized as a reasonable limit. For most time slices, the radial lines have already intersected with islands at less than 8 km distance from the coring site (see Fig. 4). As Fig. 4 illustrates, some lines will reach further before intersecting with land. With increasing length of radial lines, one or a few these far-reaching lines could contribute more to the openness index since it averages lengths for all radical lines, which may increase the relative changes of the estimated openness index through the time. We will do a sensitivity testing to quantify potential effects of the maximum length on the estimated openness index. In general, the changes caused by different lengths of radial lines will most probably not alter the trend which was detected with the current estimation.

Changes in the manuscript based on comment #3:

Page 4, line 13: The text of "The radiating lines of 8 km were used as they can reach open water region when the inlet was relatively open" is added after the sentence "The length of the radiating lines. . .was set as 1-5, 10 and 15°".

Page 5, line 10: "The shifting angles of 0° to 4° have been used to test whether different starting angles influence the openness indices. The results presented in Fig. 7 shows when using different shifting angles, the openness indices show substantial differences. However, if interval is set as 1°, changing the shifting angle from 0° to 4° would result in little differences in openness indices. Therefore using low degree interval such as 1° for calculating the openness indices is preferred and should be recommended for other similar studies, although the computing would be longer than higher degree intervals. The landward and seaward openness indices were differentiated although they both reflect morphological changes of the inlet over the last 5.4 ka. The seaward openness index better reflect the embayment process in comparison with the landward openness, as the most distinct changes of the inlet is from the sill in the east. Even so, the landward openness index, reflecting offshore distances, is calculated, though with lacking of information for the past prevailing." is added after the sentence "Therefore, the associated uncertainties. . .openness variability." The impacts of different shifting degrees on the calculated openness index will be quantified by statistics. A sensitivity test of different maximum radial lengths (10 km and 15 km) effects on the openness index will be conducted

Comment #4 The authors present Figure 6 and 7 to show there is variation in the openness index for a given degree interval (or shifting angle) at a given time. Is there a way to make this analysis more quantitative? (i.e., p5, line 9; how much 'larger'?) One suggestion to illustrate this in the figures is to plot openness variance as opposed to the raw openness index. On a related note, the authors state that they endeavor to find an optimal degree interval (p. 5 line 3). I assume 'optimal' in this context refers to a negligible variance in openness index relative to decrease computation time (associated with increasing the degree interval)? Perhaps quantifying the variation in

openness index for a given degree interval will aid them in searching for an 'optimal' interval?

Author's reply on comment #4: Different shifting angles and intervals are used to test if there is large difference among them. The results in Fig. 6 demonstrate that there are large variances among different shifting angles and intervals. We agree that it would be good to quantify the differences among different scenarios and which has now been quantified in the revised manuscript (see the changes below as well). The $1°$ interval is recognized as an 'optimal' interval in our study. Still, this is based on the fact that in our study, the computing time for using $1°$ interval is still acceptable. If further study has large data set (i.e. processing openness index focusing on many sites), the computing time may need to take into consideration and may end up with larger intervals, e.g., $2°$ or $3°$.

Changes made in the manuscript based on comment #4: Page 5, line 5: "For example, the calculated landward and seaward indices using $15°$ interval are at the maximum 7% and 20% larger than the $1°$ interval scenario." is added after the sentence "Both the seaward. . .the smaller degree intervals (Fig. 6)." Page 5, line 6: "(maximum 5%)" is added after "only minor difference".

Comment #5: The authors present openness index data and grain size in figure 8. I believe more quantitative analysis could be performed with this data to convince the readers. For instance, what values of shifting angle and degree interval was used? Why? What is the correlation between opening index vs sand %? or openness index vs silt/clay?

Authors' reply on comment #5: Shifting angle of $0°$ and interval of $1°$ are used in the Fig. 8 scenarios. When the interval of $1°$ is used, the shifting angle will only have little impact on the openness indices (see the replies on comment #3). The size of the interval indicates empties spaces without radial lines. With a reduced interval size, there is high chance to capture more detailed morphological changes and also there is

less impact from the shifting angle on the estimated index.

Changes made in the manuscript based on comment #5: To quantify the relationship between openness index with sand fraction, the R2 with significance level was calculated to better inform the readers and has now been added in the Table 1 in the revised manuscript.

Comment #6

Has there been erosion of the islands since 5 ka? (i.e., is the present subaerial expression of the islands identical to the coastlines of the island in the past?) how could this impact your study?

Author's reply on comment #6: Erosion from the islands since 5.4 ka most likely occurred but has been weak as these islands are mostly rocky. It may have caused a delivery of relatively large grains into the coring site during the land-uplift process. As the uplift process has been generally linear, we might observe a linear change in the grain size data if land uplift has played the dominant role in governing the grain size. However, the sand contents and silt/clay ratios exhibit non-linear changes, which indicate the long-term openness change of the inlet is more important.

Changes made in the manuscript based on comment #6: Page 6, line 4: "Erosion from the islands since 5.4 ka most likely occurred but has been weak as these islands are mostly rocky. It may have caused a delivery of relatively large grains into the coring site during the land-uplift process. As the uplift process has been generally linear, we might observe a linear change in the grain size data if land uplift has played the dominant role in governing the grain size. However, the sand contents and silt/clay ratios exhibit a stepwise change, which indicates the long-term openness change of the inlet is more important. Coarse grains such as sand can also be transported to the coring site through storm events, winter sea ice or drifting sea weed, although their impacts are difficult to estimate." is added after "Together with sheltered condition. . .in the sediments".
**ESurfD**

Interactive
comment

---

## Author Comment (AC2) · 1 Jul 2016

Reviewer 2's comments:

I would like to see a few more notes about the setting: are there only rocky coasts, or are there also patches of sandy shores? And what about shallow waters? All rocks? Some notes are found in 3.2, but more notes could be added to 2.1.

I also wonder how sand is transported to the core site. Does it happen during storms as storm sand layers? Is sand blown out on the sea ice during cold winters? Is sand transported by drifting sea weed or by drifting sea ice?

I would also like to see a few notes on the chronology of the core, at least a reference

to Ning et al. (2016).

The main control on grain size distribution is distance to the shore, but this is apparently not mentioned. The closer to the shore – the more coarse-grained sediments. In Gåsfjärden, however, the sediments become more and more fine-grained as the core site moves closer to the shore. This is not surprising, because the core site at the same time becomes more and more protected. The authors have developed a novel GIS-based approach that allows them to quantify down-core changes in grain size distributions in relation to changing fetch.

Author's Reply:

Thanks for the great comments. We will first give our replies to each asked question and then list all corresponding changes have made in the revised manuscript.

The shallow waters and the shore are characterized with rocky coasts and some sandy patches based on observation. Inside the inlet, there is so far no data about spatial distribution of sediment grain size. The sand content in the inlet are supposed to be relatively low, due to lack of large rivers draining into the inlet and erodible soil as well as its enclosed setting.

The sand content is generally lower than 1% in our coring site. During periods with relatively high openness, storm events would most likely transport large amount of sand and silt into the coring site which is shown in Figure 8. Sand can also be transported to the coring site through sea ice and/or drifting sea weeds, although the impacts are hard to estimate.

A description on the chronology of the core has been added.

Thanks for pointing out the underlying impacts of distances on grain size distributions. We have now addressed in the revised version.

Changes in the manuscript:

Page 3, line 15: The following descriptions have been added in the revised version. "In the shallow waters of the inlet, sandy patches and rocky coast can be found. In general, there is a lack of erodible soils and therefore subsequent sediment transportation into the inlet. The accumulated sediment in the inlet is expected to originate mostly from the terrestrial setting, compared with sediment transportation from the open Baltic Sea. Sediment accumulation rate over the last 1 ka is generally less than 1.5 mm per year in the deep basin (Ning et al., 2016)."

Page 3, line 16: "Grain-size analysis" is changed to "Chronology and grain-size analysis".

Page 3, line 18: "Age-depth model of the sediment sequence was established through a combination of 210Pb and 14C dating methods (Ning et al., 2016)." is added after "A 6 m sediment sequence was obtained covering the last 5.4 ka (Ning et al., 2016)".

Page 5, line 20: "In general, the closer to the shore, the more coarse-grained sediments will be deposited. In Gåsfjärden, however, the sediments become more and more fine-grained as the core site moves closer to the shore. This is because the core site at the same time becomes more and more protected as shown from the openness indices." is added after "We have. . .openness variations."

Page 6, line 4: "Coarse grains such as sand can also be transported to the coring site through storm events, winter sea ice or drifting sea weed, although their impacts are difficult to estimate." is added after the sentence "Coarse grains such. . .are difficult to estimate.".

Reference:

Ning, W., Ghosh, A., Jilbert, T., Slomp, C. P., Khan, M., Nyberg, J., Conley, D. J., and Filipsson, H. L.: Evolving coastal character of a Baltic Sea inlet during the Holocene shoreline regression: impact on coastal zone hypoxia, J Paleolimnol, 55, 319-338, 10.1007/s10933-016-9882-6, 2016.

**ESurfD**

Interactive
comment

---

## Author Response (AR1)

The authors would like to thank reviewer Evan Goldstein for giving the constructive comments, which will definitely improve the manuscript. Below we firstly addressed each comment and also indicate changes that have been made in the revised manuscript. Notably, the page and line numbers are the ones in the attached revised manuscript.

**Comment #1:**

I believe this manuscript could benefit with more description as to the mechanics of sediment transport in this specific system to justify the results (Section 3.2). For instance, what drives sand transport in the modern system? Does sand come from the Baltic into the inlet? Or is the sand coming from the terrestrial setting? i.e., as a reader it would be helpful to understand in more detail how this physical system works?

**Authors' reply on comment #1:**

Thanks for the good points. As the catchment of the inlet is characterized with thin soil and the inlet only has a few small rivers draining into it. On one hand, we speculated that sand transportation into the inlet from the catchment could be limited. On the other hand, the sand input into the inlet from the offshore regions could be very small, as a result of the narrow and shallow sill between the inlet and the open sea water. Overall, sand and/or sediment can be transported into the inlet through both terrestrial input and offshore region with limited amount (supported by relatively low sedimentation rate, <1.5 mm yr$^{-1}$, during the last 1 ka). We speculated that sediment accumulated in the inlet mostly originates from the terrestrial setting, compared with the sandy offshore region.

**Changes made in the manuscript based on comment #1:**

Page 4, line 13: "which hinders sediment transportation between the inlet and the open water." is added after the sentence "It has a restricted water exchange with the open Baltic Sea through a narrow and shallow strait (500 m wide, <20 m deep) in the east".

Page 4, line 20: "In the shallow waters of the inlet, sandy patches can be found in addition to the rocky coast. The sediment accumulating in the inlet most likely originates from the terrestrial environment through erosion, and from land-run and river transport, instead from the open Baltic Sea. Sediment accumulation rate over the last 1000 years is generally less than

1.5 mm yr-1 in the deep basin (Ning et al., 2016)." is added after the sentence "The RSL has decreased by 17 m in…1.5 mm yr$^{-1}$".

**Comment #2:**

Can the authors connect openness index with a near bottom water velocity and sediment transport in some way - i.e., fetch, wind speed, and water depth to calculate wave orbital motions at the bed using the relations presented in Young and Verhagen (1996)? Or perhaps the authors could relate the (spatial) change in openness index to the wind field (modern or ancient) and the fetch?

**Author's Reply on comment #2:**

Thanks for the great point. It would be really interesting to relate the estimated openness index with other environmental variables to potentially explain sediment transport mechanism. However, there is a lack of reliable data on historic wind speed and direction in the Baltic Sea region. Thus, calculating wave orbital motions at the bed for the long-term scale is not possible at this moment.

As the reviewer correctly pointed out, it could be interesting to link the spatial change in the openness to the changes of wind field in the inlet, which may improve the ability of our current method in explaining sediment variation. However, since the main focus of this study is to explain the temporal dynamics of grain size changes at long-term scale, it may bring additional uncertainties to our estimations if we only use the available modern wind data for the past 5.4 ka. With availability of reliable wind data at long-term scale in future, it will definitely be interesting to explore potential impacts of wind on the grain size changes at the core site.

**Changes made in the manuscript based on comment #2:**

Page 8, line 18, some discussion about landward openness index with wind direction has been added.

**Comment #3:**

The authors focus on developing an 'openness index' which is the average length of line from the core site to land at a given time/sea level. Why are landward vs. seaward openness indices differentiated? And a related comment, the shifting angle

is discussed only briefly. Can the authors give us some guidance on picking a starting position? Do any radial lines, at any time, make it to the open Baltic sea (i.e., do any openness measurements exceed the 8 km line segments used)? Are these lines important? (I would presume so, because these directions would permit larger waves into the system and exert more work on the bed.)

**Author's reply on comment #3:**

Both the seaward and landward openness indices can be linked with fetch and wave energy in the inlet, where high indices values potentially indicate relatively large bottom velocity. Thus higher openness indices in the open system lead to larger grain size in the sedimentation area. In comparison with the landward openness, the seaward openness index better reflects the morphological changes of the inlet, which is the main cause for hydrodynamic energy changes in the inlet over the last 5.4 ka. The landward index is used to describe the changes in offshore distances and it can be important if prevailing wind direction is from the land to the sea.

Thanks for the suggestions on adding comments on the shifting angle. The shifting angles of 0° to 4° have been used to test whether different starting angles influence the openness indices. The results presented in Fig. 7 showed that using 5° interval and different shifting angles, the changes in openness indices were substantial. If the interval is set as 1°, changing the shifting angle from 0° to 4° would be lead to little changes in the openness indices. Therefore using low degree interval such as 1° for calculating the openness indices is preferred. The optimal interval for estimating openness index could vary from different coastal settings and we suggest to test it before apply the index with other proxy data. In our study, using 1° interval would give the most robust results when calculating openness indices, although the computing time would be longer than larger degree intervals.

We have sediment data from the core site and this is also the site we are interested to investigate factors impacting sedimentation process. So it is straightforward for us to use the core site as the starting point of radial lines and the estimated changes of openness index can further link the index with other measured sediment variables.

The maximum length of 8 km line was used because it reached open water region for scenarios at the 5.4 ka ago and is recognized as a reasonable limit. For most time slices, the

radial lines have already intersected with islands at less than 8 km distance from the coring site (see Fig. 4). As Fig. 4 illustrates, some lines will reach further before intersecting with land. With increasing length of radial lines, one or a few these far-reaching lines could contribute relatively more to the estimated openness index, which may increase the relative changes of the estimated openness index through the time. However, we think the changes caused by different lengths of radial lines will most probably not alter the trend which was detected with the current estimation.

**Changes in the manuscript based on comment #3:**

Page 6, line 18: The text of "The radial lines of 8 km were used as they can reach offshore open water region" is added after the sentence "The length of the radiating lines…was set as 1-5, 10 and 15°".

Page 8, line 5: ".Furthermore, potential effects of shifting angles (angle between the north and the nearest radial line) were tested and Figure 7 showed the cases for 5° intervals of radial lines with shifting angles of 0° to 4°. The results demonstrated that using different shifting angles can cause substantial differences in the estimated openness indices when the radial intervals are relatively large. However, if the interval is set as 1°, changing the shifting angle from 0° to 4° would result in openness indices with little very small differences in consideration of relative changes in the positions of all radial lines. Therefore using low degree interval such as 1° for calculating the openness indices is preferred and should be recommended for other similar studies, although the computing time would be longer than higher degree intervals. The landward and seaward openness indices were differentiated although they both reflect morphological changes of the inlet over the last 5.4 ka. The seaward openness index reflected more accurately the embayment process in comparison with the landward openness index, as the most distinct changes of the inlet was from the sill in the east. The landward openness index, reflecting offshore distance, could also be important for considering sedimentary grain size, especially if information describing for the past prevailing wind direction becomes available." is added after the sentence "Therefore, the associated uncertainties…openness variability."

**Comment #4**

The authors present Figure 6 and 7 to show there is variation in the openness index for a given degree interval (or shifting angle) at a given time. Is there a way to make this analysis more quantitative? (i.e., p5, line 9; how much 'larger'?) One suggestion to illustrate this in the figures is to plot openness variance as opposed to the raw openness index. On a related note, the authors state that they endeavor to find an optimal degree interval (p. 5 line 3). I assume 'optimal' in this context refers to a negligible variance in openness index relative to decrease computation time (associated with increasing the degree interval)? Perhaps quantifying the variation in openness index for a given degree interval will aid them in searching for an 'optimal' interval?

**Author's reply on comment #4:**
Different shifting angles and intervals are used to test if there is large difference among them. The results in Fig. 6 and 7 demonstrate that there are large variances among different shifting angles and intervals. We agree that it would be good to quantify the differences among different scenarios and which has now been quantified in the revised manuscript (see the changes below as well). The 1° interval is recognized as an 'optimal' interval in our study, which is based on the fact that the computing time for using 1° interval is still acceptable. If further study has large data set (i.e. processing openness index focusing on many sites), the computing time may need to take into consideration and may end up with larger intervals, e.g., 2° or 3°.

**Changes made in the manuscript based on comment #4:**
Page 7, line : "The calculated landward and seaward indices using 15° interval are at the maximum 7 % and 20 % larger than the 1° interval scenario." is added after the sentence "Both the seaward…the smaller degree intervals (Fig. 6)."
Page 7, line 18: "(maximum 5 %)" is added after "only minor difference".

**Comment #5:**
The authors present openness index data and grain size in figure 8. I believe more quantitative analysis could be performed with this data to convince the readers. For instance, what values of shifting angle and degree interval was used? Why? What is the correlation between opening index vs sand %? or openness index vs silt/clay?

**Authors' reply on comment #5:**

Shifting angle of 0° and interval of 1° are used in the Fig. 8 scenarios. When the interval of 1° is used, the shifting angle will only have little impact on the openness indices (see the replies on comment #3). The size of the interval indicates empties spaces without radial lines. With a reduced interval size, there is high chance to capture more detailed morphological changes and also there is less impact from the shifting angle on the estimated index.

**Changes made in the manuscript based on comment #5:**

To quantify the relationship between openness index with sand fraction, the $R^2$ with significance level was calculated to better inform the readers and has now been added in the Table 1 in the revised manuscript.

**Comment #6**

Has there been erosion of the islands since 5 ka? (i.e., is the present subaerial expression of the islands identical to the coastlines of the island in the past?) how could this impact your study?

**Author's reply on comment #6:**

Erosion from the islands since 5.4 ka most likely occurred but has been weak as these islands are mostly rocky. It may have caused a delivery of relatively large grains into the coring site during the land-uplift process. As the uplift process has been generally linear, we might expect to see a linear change in the grain size data if land uplift has played the dominant role in governing the grain size. However, the sand contents and silt/clay ratios exhibit non-linear changes, which indicate other factors than land uplifting could also participate in influencing grain-size distribution.

**Changes made in the manuscript based on comment #6:**

Page 10, line 3: "Erosion from the surrounding islands since 5.4 ka has most likely occurred, but could be rather limited as these islands are mostly rocky with little soil cover. It may, however, result in a flux of relatively coarser grains to the coring site during the land-uplift. As the uplift process close to be linear (see Fig. 2), we might expect to see a rather linear change in the grain size data assuming the land uplift played the dominant role. However, the sand content and silt/clay ratios exhibit strong year-to-year variations, which indicates other factors than land uplifting could also participate in influencing grain-size distribution. For instance, coarse grains, such as sand, can also be transported to the coring site through storm

events and intense wave action, sea ice or drifting sea weed. However, their impacts are not explicitly included in the openness indices." is added after "Together with sheltered condition…in the sediments".

**Replies to the 2nd reviewer:**

I would like to see a few more notes about the setting: are there only rocky coasts, or are there also patches of sandy shores? And what about shallow waters? All rocks? Some notes are found in 3.2, but more notes could be added to 2.1.

I also wonder how sand is transported to the core site. Does it happen during storms as storm sand layers? Is sand blown out on the sea ice during cold winters? Is sand transported by drifting sea weed or by drifting sea ice?

I would also like to see a few notes on the chronology of the core, at least a reference to Ning et al. (2016).

The main control on grain size distribution is distance to the shore, but this is apparently not mentioned. The closer to the shore – the more coarse-grained sediments. In Gåsfjärden, however, the sediments become more and more fine-grained as the core site moves closer to the shore. This is not surprising, because the core site at the same time becomes more and more protected. The authors have developed a novel GIS-based approach that allows them to quantify down-core changes in grain size distributions in relation to changing fetch.

**Author's Reply:**

Thanks for the great comments. We will first give our replies to each asked question and then list all corresponding changes have made in the revised manuscript. Notably, the page and line numbers are the ones in the attached revised manuscript.

The shallow waters and the shore are characterized with rocky coasts and some sandy patches based on observation. Inside the inlet, there is so far no data about spatial distribution of sediment grain size. The sand content in the inlet are supposed to be relatively low, due to lack of large rivers draining into the inlet and erodible soil as well as its enclosed setting.

The sand content is generally lower than 1% in our coring site. During periods with relatively high openness, storm events would most likely transport large amount of sand and silt into the coring site which is shown in Figure 8. Sand can also be transported to the coring site through sea ice and/or drifting sea weeds, although the impacts are hard to estimate.

A description on the chronology of the core has been added.

Thanks for pointing out the underlying impacts of distances on grain size distributions. We have now addressed in the revised version.

**Changes in the manuscript:**

Page 4, line 20: In the shallow waters of the inlet, sandy patches can be found in addition to the rocky coast. The sediment accumulating in the inlet most likely originates from the terrestrial environment through erosion, and from land-run and river transport, instead from the open Baltic Sea. Sediment accumulation rate over the last 1000 years is generally less than 1.5 mm yr-1 in the deep basin (Ning et al., 2016)."

Page 5, line 16: "Grain-size analysis" is changed to "Chronology and grain-size analysis".

Page 5, line 18: "and the age-depth model of the sediment sequence was established through a combination of $^{210}$Pb and $^{14}$Cs and AMS-$^{14}$C dating methods (Ning et al., 2016)." is added after "A 6 m sediment sequence was obtained covering the last 5.4 ka (Ning et al., 2016)".

Page 9, line 8: "The maximum sand content at the core site was only 0.4%, suggesting a relatively low bottom water velocity compared with open Baltic Sea waters (Jönsson et al., 2005). Generally, the closer to the shore, the more coarse-grained sediments can be deposited. However, the sediments in Gåsfjärden became more and more fine-grained as the coring site became shallower and closer to the shore (closer to present time, see Figs. 5 and 8), which was a result of less exposure and an increasingly protected location (reflected by the openness index).." is added after "We have…openness variations."

[revised manuscript text omitted]

---

## Author Response (AR2)

Dear Editor,

Thanks for the detailed correction for this manuscript. We have now corrected all suggested changes.

In the text you are using 'ka', 'ka PB', 'yr', 'year', 'years', and in the figures 'Year (ka BP)' as axes label, and 'ka' as well as 'ka BP' and 'ka years' within the figures.

The use of different abbreviations has now been corrected in the main text as well as in figures (caption).

Please be consistent throughout, us 'a' [annum] as unit for time and 'ka' where appropriate. Please refrain from using 'BP'. This should be reserved for uncalibrated (and thus wrong) radiocarbon years. Please replace 'year' as axes label with 'age'.

Thanks for pointing out. We have now corrected figures with right axes labels.

Figure 1; Figure caption b): replace 'vegetation' with 'land use'; 'in the study region' is needed only once

Corrected.

Figure 2; Figure caption: Reword – you show the relative sea level not its variation

Corrected.

Figure 3: change 'radical lines' to 'radial lines'

Corrected

In Figure 8 please plot data points only and do not connect points with a line. Reword the figure caption: you do not show a comparison – you just show 4 different graphs next to each other

Corrected.